# Potential of Porous Substrate Bioreactors for Removal of Pollutants from Wastewater Using Microalgae

**DOI:** 10.3390/bioengineering10101173

**Published:** 2023-10-09

**Authors:** Dora Allegra Carbone, Michael Melkonian

**Affiliations:** 1Laboratory of Biological Oceanography, Stazione Zoologica “A. Dohrn”, Villa Comunale, 80121 Naples, Italy; 2Integrative Bioinformatics, Department of Plant Microbe Interactions, Max Planck Institute for Plant Breeding Research, Carl-von-Linne-Weg 10, 50829 Cologne, Germany

**Keywords:** microalgae, wastewater, porous substrate bioreactor, pollutants removal, mass cultivation

## Abstract

Porous substrate bioreactors (PSBRs) are a new technology to grow microalgae immobilized in a dense culture and solve some problems linked to suspended cultivation. During recent years, this technology has been used in laboratory and pilot setups in different fields of environmental biotechnology, such as wastewater treatment. The aim of this short review is to introduce the PSBR technology, summarize the results obtained in removing some pollutants from wastewater, provide an assessment of the potential of PSBRs for wastewater treatment, and the subsequent use of the algal biomass for other purposes.

## 1. Introduction

Freshwater, and in particular drinking water, scarcity is becoming a threat to sustainable development of human society. It has been estimated that two-thirds of the global population live under conditions of severe water scarcity for at least one month of the year, and half a billion people in the world face severe water scarcity throughout the whole year [1]. Global water scarcity is driven by both water quantity and water quality issues, requiring expansions in clean water technologies, e.g., wastewater (WW) reuse, to achieve Sustainable Development Goal 6 of the UN [2]. Freshwater quality can be compromised by a multitude of pollutants including heavy metals, nutrients (nitrogen and phosphorus), organic substances, pathogens including viruses, acidification, etc. These factors make the water unusable [3,4] and, consequently, WW treatment has become one of the principal current environmental challenges [5,6]. In addition, some essential elements for the existence of life on earth, such as phosphorus, are rapidly lost from agricultural soils due to erosion, limiting food and feed production and requiring recovery from WW [7].

Conventional treatment methods for WW are generally characterized by high costs of operation and removal of the sludge produced [8]. Bioremediation of WW using microalgae has gained increased attention in recent years, being more eco-friendly and possibly more cost-effective than conventional WW treatment processes [9,10,11]. The most common method of microalgae cultivation for WW treatment is the suspended system, in which organisms grow suspended in water in open ponds or photobioreactors [12,13]. The high-rate algal ponds (HRAPs) were introduced by Oswald and Gotaas [14] in California (see recent reviews in [15,16]). HRAPs have been used for different types of WW such as domestic, tannery, dairy, and piggery. Although progress has been made in recent years to optimize design and performance of HRAPs, they still exhibit low productivity, requiring mixing (with paddle wheels) and addition of carbon dioxide (since they are carbon-limited). HRAPs are difficult to maintain (in particular, as monocultures), because of their exposure to predators, parasites, and competitors. Harvesting microalgae from low density suspensions is cost-intensive, negatively impacting the implementation of circular economy (CE) principles.

Suspended cultivation of algae in closed photobioreactors has shown some advantages over HRAPs, such as reduced contamination, higher levels of biomass, faster growth, and better control of selected parameters (light, temperature, pH). Closed photobioreactors, however, require large investments for set-up, high operational costs for mass transfer and for harvest, and are thus not economically sustainable for WW treatment. More recently, cultivation methods involving immobilization of microalgae have gained increased attention [17,18,19]. In immobilized systems, biomass and the bulk of the liquid medium are separated; consequently, there is less consumption of energy linked to mixing and harvesting, because the biomass is harvested directly as fresh weight [17,20,21,22,23,24].

Algal cell immobilization can be obtained by various means but one of the most widely used immobilization systems is entrapment of cells in a three-dimensional gel lattice (beads) using either synthetic (e.g., polyurethane) or natural (e.g., alginate, carrageenan) polymers (see a review by Mallick [25]). This type of system has several disadvantages including creation of secondary pollution linked to the presence of the polymeric substance, being cost-intensive, and resulting in significant leakage of microalgae from the gel matrix, especially during long-term cultivation [26,27,28].

Another commonly used method for algal cell immobilization in WW treatment is the creation of a biofilm [29]. There is a growing interest in biofilm-based algal cultivation systems, because they could provide a more cost- and energy-efficient alternative to suspended cultivation systems [30]. In the most widely used technical set-up, the algal turf scrubber (ATS) system [31], a natural assemblage of periphyton (mostly filamentous algae) self-immobilizes on a submerged substrate. WW flows in a series of pulses along an inclined flow way over the biofilm [31]. The biomass is harvested by periodical “scrubbing” from the substrate. This and other biofilm systems that are at least intermittently submerged in WW (such as rotating biological contractors, [32]), although simple to operate, display some drawbacks. There is little control over the composition of the periphyton community that develops on the biofilm. A significant component of the biofilm consists of heterotrophic organisms such as bacteria, protists, and fungi, as well as detritus. The biomass needs to be harvested frequently to minimize biomass leakage and slowdown of growth. Finally, exposure to predators and parasites cannot be prevented. In general, in submerged phototrophic biofilms, light, carbon dioxide, and nutrients all pass through a layer of (often turbid) WW before reaching the biofilm surface, being attenuated on the way. A graphical overview of the different biofilm systems in use is presented in Figure 1.

A new type of immobilized algal cultivation system, largely avoiding these problems, was introduced in the early 2000s, and later generalized under the term porous substrate bioreactors (PSBRs, [34]). Initially developed for use in long-term biosensors to detect volatile toxic compounds [35], the central component of this immobilization system is a hydrophilic, microporous layer (often a membrane) that separates the algal biofilm from the bulk of the liquid (WW, culture medium). The biofilm is immobilized on one surface of the layer, while the liquid medium is confined to the opposite surface and is usually moved unidirectionally along this surface. The microporous layer with its two surfaces has been termed the twin-layer [36]. In practice, to regulate the flow of the liquid medium and to provide a sufficient source of the liquid, an additional hydrophilic, macroporous layer has been incorporated into the system, with the two layers constituting the substrate layer (the microporous layer carrying the biofilm) and the source layer (the macroporous layer holding the liquid medium) [36,37]. The advantages of porous substrate bioreactors over the other immobilization systems, described above, are easy to understand: (1) contamination in this open system is minimized because the microporous (usually <1 µm pore size) layer prevents access to the biofilm and spreading of predators, pathogens, and parasites, (2) leakage of algae from the biofilm is prevented because of the strict separation of the biofilm from the flowing liquid to the two surfaces of the twin-layer (TW-S), (3) since the biofilm is directly exposed to the atmosphere, the diffusion paths of carbon dioxide and light to the biofilm (and oxygen from the biofilm) are extremely short, resulting in relatively high biomass productivities, and (4) PSBRs allow the cultivation of microalgae as monocultures, supporting product-based applications in microalgal biotechnology [21].

Porous substrate bioreactors have also been described as “attached cultivation technology” (A-T) [38,39]. The use of the term “attached cultivation” for PSBRs, however, should be discouraged, because it is ambiguous, referring also to submerged biofilms [40]. Algal PSBRs have been reviewed on several occasions in recent years, with emphasis on the mechanisms of growth and applications in biotechnology [17,18,21,41,42]. In this article, we review recent progress in the removal of pollutants from wastewater using algal PSBRs and the potential use of the microalgal biomass obtained during this process.

## 2. General Facts about the Use of PSBRs in the Removal of Pollutants from Wastewater

Generally, in bench scale experiments, the PSBRs are placed vertically inside transparent material (plastic tube or glass plate). The WW medium is applied to the top of the source layer by means of a peristaltic pump with a flow speed of 6–8 mL min^−1^ [37,38,43]. The source and substrate layers consist of different materials, preferably synthetic (non-woven fabric, polycarbonate, nylon, or nitrocellulose membranes), to avoid the development of fungi [37]. To minimize clogging of the source layer, only settled WW is used [44]. Microalgae used for the experiments are generally selected based on their tolerance against diverse pollutants, and thus have often been isolated from polluted environments [45]. Moreover, until now, only green microalgae (Chlorophyta) have been used on PSBRs for removal of pollutants from WW, and generally the duration of experiments has not exceeded three months [16,46]. This choice is supported by literature data [47] and some experimental tests. In their work, Gonzales et al. [16] isolated 33 different algal strains from WW and then tested microalgae directly on TW-S for 12 days in presence of wastewater. They showed that the green microalga *Scenedesmus* sp. achieved the highest biomass (12 g m^−2^), whereas the cyanobacterium *Phormidium* sp. exhibited the lowest (3 g m^−2^).

## 3. Removal of Pollutants from WW

The origin of WW is variable (domestic, industrial, infiltration, storm water), and consequently its chemical composition also differs [44]. However, WW, in general, is often rich in nutrients such as nitrogen, phosphorus, and chemical oxygen demand (COD); these are the main cause of eutrophication of water bodies, leading to excessive production of organic matter and causing damage to ecosystems and their biota [48,49]. Bioremediation of eutrophic water bodies is not simple: for example, phosphorus removal often involves the supply of chemical reagents and the production of high quantities of chemical sludge; nitrogen removal is complex, requiring alternating cycles of denitrification and nitrification, and is both energy-intensive and incomplete [50]. Several studies have shown that microalgae are one of the most promising candidates for the phytoremediation of nitrogen and phosphorus, as well as some organic compounds from different water bodies, saving costs and energy [51].

The first experiment with WW using a TW-S was carried out by Shi et al. [37]. *Scenedesmus* (*Halochlorella*) *rubescens* and *Chlorella vulgaris* were tested in a greenhouse from April to July under ambient illumination from 20 to 120 μmol photons m^−2^ s^−1^ in batch mode, in the presence of municipal wastewater (MWW) or synthetic wastewater (SWW). The MWW was collected from Stadtentwässerungsbetriebe Köln (Cologne, Germany) after nitrification and denitrification and treatment in an aerobic activated sludge system [44]. The SWW was a modified BG11 culture medium with a higher content of phosphate (KH_2_PO_4_), nitrate (NaNO_3_), and ammonium (NH_4_Cl). In the MWW experiment, after immobilization on the TW-S, the two microalgae grew for eleven days in normal BG11 medium, then three days in a medium without nitrate, and finally in MWW for seven days; in the SWW experiment, microalgae grew for nine days without preadaptation in SWW. In MWW, initially, nitrate was 6.2 mg L^−1^ (Table 1) and it was removed within three days after a lag phase of one day with a removal efficiency of 93%, achieving values of 0.3 mg L^−1^ (Figure 2). After a replenishment of MWW, microalgae removed nitrate within two days with the same efficiency and without a lag phase.

**Table 1 bioengineering-10-01173-t001:** Removal of inorganic compounds (ammonium, phosphorus, nitrate) and COD from specific wastewaters by microalgae using PSBRs.

		Ammonium			Phosphorus			
Microalgae	Medium/ Substrate	Initial Value mg L^−1^	Final Value mg L^−1^	Final Removal Efficiency %	Initial Value mg L^−1^	Final Value mg L^−1^	Final Removal Efficiency %	References
*Scenedesmus (Halochlorella) rubescens*	SWW	20	1.16	94	3	0.2	93	Shi et al. [37]
*Chlorella vulgaris*	SWW	20	0.87	96	3	0.1	96	Shi et al. [37]
*Desmodesmus abundans*	Human urine	5.76	4.8	13	0.29	0.02	94	Piltz and Melkonian [52]
*Chlorella pyrenoidosa*	SWIW	409	98	76	35	10	71	Cheng et al. [43]
*Scenedesmus* sp.	SYSWIW + antibiotics	50	9	83	/	/	/	Cheng et al. [53]
*Chlorella* sp.	ADSW-5D	134	2.67	94	6.65	1.05	84.3	Cheng et al. [54]
*Scenedesmus* sp.	RWW	47.04	12	85	5.08	1.88	97	Saleem et al. [55]
*Scenedesmus* sp.	WTWW	25	0	80	1.70	0.3	83	Saleem et al. [55]
*Tetradesmus obliquus*	PWW	110	0	98	12.56	1	27	Sohail et al. [56]
*Tetradesmus obliquus*	AD	104	0	90	10.61	0.5	16	Sohail et al. [56]
*Tetradesmus obliquus*	SWL	104	0	93	16.93	9	2.4	Sohail et al. [56]
*Tetradesmus obliquus*	MWW	47	0.3	97	9.07	0	100	Sohail et al. [56]
*P. maculatum*	PRNM	14.02	0.29	97	8.84	1.5	97	Meril et al. [57]
*P. maculatum*	PP	14.02	3	78	8.84	2.77	69	Meril et al. [57]
*P.maculatum*	NM	14.02	2	85	8.84	2	77	Meril et al. [57]
		**Nitrate**			**COD**			
*Scenedesmus (Halochlorella) rubescens*	SWW	3	0.2	96	/	/	/	Shi et al. [37]
*Chlorella vulgaris*	SWW	3	0.2	96	/	/	/	Shi et al. [37]
*Scenedesmus (Halochlorella) rubescens*	MWW	6.2	0.11	98	/	/	/	Shi et al. [37]
*Chlorella vulgaris*	MWW	6.2	0.22	96	/	/	/	Shi et al. [37]
*Chlorella pyrenoidosa*	SWIW	/	/	/	601	152	74	Cheng et al. [43]
*Chlorella* sp.	ADSW-5D	14.3	2.29	85.5	116	16.12	86.8	Cheng et al. [54]
*Scenedesmus* sp.	RWW	0.0287	0.0	100	458	350	25	Saleem et al. [55]
*Scenedesmus* sp.	WTWW	1.73	0.7	70	296	180	40	Saleem et al. [55]
*P. maculatum*	PRNM	6.09	0.45	92	/	/	/	Meril et al. [57]
*P. maculatum*	PP	6.09	1.45	0.76	/	/	/	Meril et al. [57]
*P. maculatum*	NM	6.09	0.8	86	/	/	/	Meril et al. [57]

**Figure 2 bioengineering-10-01173-f002:**
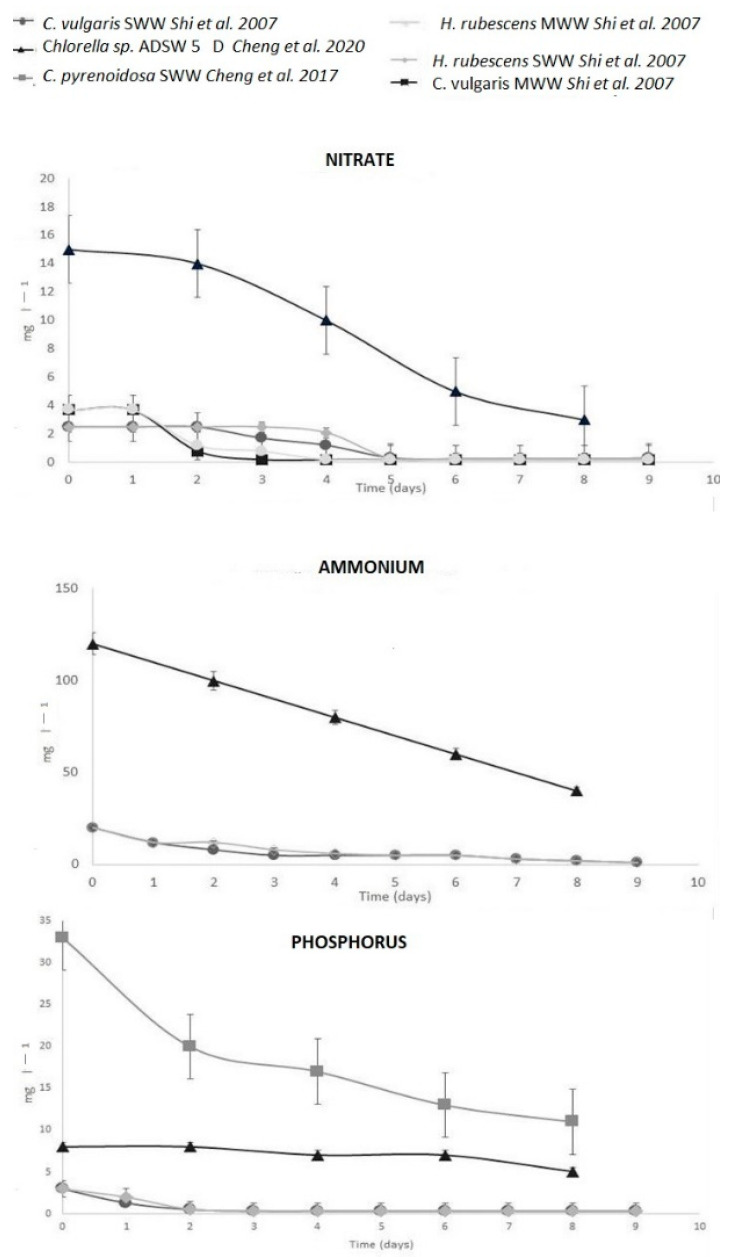
Removal trends of ammonium, nitrate, and phosphorus in different PSBR experiments of Shi et al. 2007 [37], Cheng et al. 2007 [43], and Cheng et al. 2020 [54].

In a recent study about bioremediation of MWW [55], the authors tested *Scenedesmus* sp. on real raw municipal wastewater (RWW) and wetland-treated municipal wastewater (WTWW). The experiment was set up in a horizontal TW-S at a continuous light intensity of 200 μmol photons m^−2^ s^−1^ with atmospheric air supplied by a pump for four days. There were different nutrient removal efficiencies for both media (nitrate ~100% in RWW and ~70% in WTWW, whereas ammonium removal efficiencies were ~85% in RWW and ~95% in WTWW; Table 1) but the final biomass level was 11% higher in RWW (Table 2). In the SWW experiment of Shi et al. [37], phosphate was removed within two days with a removal efficiency of around 80%, reaching values of 0.6 mg L^−1^ (Figure 2). At the beginning of the experiment, the ammonium values were 20 mg L^−1^, and at the end of the experiment they were 0.9 and 1.2 mg L^−1^ for *C. vulgaris* and *S. rubescens*, respectively (Table 1). Nitrate was removed after a lag phase of two days for *C. vulgaris* and four days for *S. rubescens* with a similar final removal efficiency (Figure 2, Table 1). At the end of the experiment, *C. vulgaris* achieved biomass values of 11.9 g m^−2^, and *S. rubescens* of 7.7 g m^−2^ (Table 2).

A TW-S was also used to investigate the removal efficiency of ammonium and phosphorus from source-separated human urine [52]. Urine was taken from student volunteers. In a preliminary test, the authors tested 96 microalgae in microtiter plates in the presence of different urine concentrations and showed that nine green microalgal strains (Chlorophyta) had the best growth performance. The nine microalgae were tested in a TW-S in the presence of 2.5% (*v*/*v*) of CO_2_ and at a light intensity of 600 μmol photons m^−2^ s^−1^ in a semi-continuous mode with medium exchange every three days in the presence of urine, diluted 1:1 and pre-treated with activated carbon to eliminate potentially detrimental effects of pharmaceuticals. Among the tested strains, only *Desmodesmus abundans* (strain CCAC 3496) showed linear growth for a period of nine days in diluted urine. Therefore, this microalga was selected for the analysis of the removal efficiency. After five days, the removal efficiency of ammonium was 13%, whereas the removal efficiency of phosphorus was 94% (Table 1), and a relatively high biomass level (75 g m^−2^) was obtained after nine days of cultivation (Table 2).

A-T was used to treat swine wastewater (SWIW), containing very high levels of pollutants such as organic substances and antibiotics, to test the removal efficiency of *Chlorella pyrenoidosa* in PSBRs [43]. The experiment was set up in non-diluted SWIW from a private farm in Wuhan city, Hubei province (China), after a preliminary treatment. *C. pyrenoidosa* was grown in laboratory conditions at a continuous light intensity of 100 μmol photons m^−2^s^−1^ and atmospheric air in batch mode for eight days. The authors monitored algal growth for eight days and reported final biomass values of 48 g m^−2^ (Table 2). They also determined the removal efficiency of total phosphorus, ammonium, and COD (Table 1, Figure 2). In 2020, the same authors [53] also analyzed growth of *Scenedesmus* sp. in same light conditions and at the same time intervals. Microalgae were grown at different concentrations of NH_4_Cl and antibiotics to create a synthetic swine wastewater (SYSWIW). Antibiotics were introduced, because swine wastewater (SWIW) often contains them. Indeed, the massive, often prophylactic, use of antibiotics in different applications (medical, veterinary, agriculture) creates a WW that spreads antibiotic resistance in bacteria in the natural environment [58,59], and different studies have shown that microalgae can absorb antibiotics in their biomass [60,61]. The different concentrations of NH_4_Cl used (in BG11 culture medium) by Cheng et al. [53] were 50 mg L^−1^, 500 mg L^−1^, and 2000 mg L^−1^. *Scenedesmus* sp. showed the best growth performance at 50 mg L^−1^ NH_4_Cl, reaching a biomass level of 55 g m^−2^ (Table 3), and the worst performance at 2000 mg L^−1^ NH_4_Cl, with a biomass level of 25 g m^2^ (Table 2) The antibiotics used for the experiment were tetracycline, sulfadimidine, and norfloxacin at different concentrations (5, 10, 20, 100, 200 mg L^−1^). *Scenedesmus* sp. grew in the presence of all three antibiotics and achieved maximum biomass levels, that were comparable to the control (Table 2), in the presence of tetracycline, at a concentration of up to 20 mg L^−1^, in the presence of norfloxacin of up to 10 mg L^−1^, and in the presence of sulfadimidine, of up to 100 mg L^−1^ (60 g m^−2^, 58 g m^−2^, 55 g m^−2^ respectively, Table 2). Thus, the removal efficiency in a SYSWIW with the optimized concentrations of ammonium and antibiotics (see above) was tested. After ten days, the removal efficiency was ~83% for ammonium (Table 1), 75% for tetracycline, 70% for norfloxacin, and 63% for sulfadimidine [53].

Cheng et al. [54] continued experiments with A-T using *Chlorella* sp., after different strains had been tested in suspended cultivation in the presence of anaerobically digested swine wastewater (ADSW) from the Yiwang livestock and poultry breeding company in Wuhan City, China. *Chlorella* sp. grew in the presence of ambient air at a light intensity of 80 μmol photons m^−2^ s^−1^ in batch mode for twelve days, in the presence of four different ADSW concentrations (the original WW, and a 2-fold, 5-fold, and 10-fold dilution). The microalgae showed the lowest biomass level (around 10 g m^−2^) in the original ADSW, and the highest biomass level at a 5-fold dilution (45 g m^−2^, Table 2). At this dilution, microalgae showed the best removal efficiency of COD and also of ammonium, nitrate, and total phosphorus (Table 1, Figure 2).

The TW-S has also been used for the treatment of aquaculture effluent [57]. In this study, the green microalga *Picochlorum maculatum* grew for 15 days in shrimp culture effluent (collected from a semi-intensive pond system at Mallipatinam (India) at 50 μmoL photons m^−2^ s^−1^ using a light/dark cycle of 12:12h and atmospheric air). Different types of substrate layers were tested (Protran reinforced nitrocellulose membrane (PRNM), nylon mesh (NM), and printing paper (PP)). *P. maculatum* showed the highest biomass levels on PNMR (around 15 g m^−2^), and also the maximum removal efficiency of different nutrients (e.g., ammonium 97%). The lowest removal efficiencies and biomass levels were observed on PP (ammonium 78%; biomass level 3 g m^−2^).

Finally, Sohail et al. [56] tested the removal of pollutants from different high-nutrient WW (solid waste leachate (SWL), poultry wastewater (PWW), and anaerobic digestate (AD)), diluted with wetland-treated municipal wastewater (MWW) by the green microalga *Tetradesmus obliquus*, using a TW-S. The dilutions kept the initial NH_4_ ^+^-N values at around 100 mg L^−1^ (higher levels are often toxic to microalgae). The experiment was set up for 18 days with provision of ambient air at a light intensity of 80 μmol photons m^−2^ s^−1^ and a light/dark cycle of 14:10 h. Moreover, in this experiment, the TW-S was covered by melamine to minimize contamination. In all types of diluted high-nutrient WW, growth and removal efficiencies for ammonium and phosphorus were significant (Table 1), but PWW exhibited maximum removal rates per unit area and the highest biomass productivity (Table 2).

## 4. Removal of Heavy Metals

Heavy metals are discharged in water by different types of chemical or mining industries. These substances can have high solubility in the aquatic environment, can be absorbed by different types of organisms, and accumulate in the food web, causing significant health problems [62,63]. Bioremediation of heavy metals using microalgae has long been regarded as the gold standard for metal detoxification of surface waters [64,65,66] because of the negative surface charges of algal cell walls [67]. In some details, the bioremediation process of heavy metals in microalgae is biphasic. First, there is passive adsorption of metals by extracellular polysaccharides and other cell wall components during the first few hours. This is followed by active uptake of heavy metals into the cell with subsequent detoxification by various mechanisms [68]. High heavy metal removal efficiencies thus depend both on the composition and thickness of the algal cell wall, as well as on uptake and detoxification mechanisms for the particular metal species. The latter is dependent on the physiological state of the organism, which in turn is influenced by growth-relevant parameters such as light, nutrients and temperature, and must be determined empirically.

Heavy metal bioremediation by microalgae has also been analyzed using PSBRs [69,70]. Cheng et al. [69] used A-T to test the ability of *Botryococcus braunii* to survive in the presence of cobalt at six different concentrations (0.09, 0.18, 0.45, 0.90, 4.5, and 45 mg L^−1^) in batch condition at a continuous light intensity of 100 μmol photons m^−2^ s^−1^ in presence of 1% (*v*/*v*) CO_2_. The highest cobalt concentration applied (45 mg L^−1^) was toxic for *B. braunii*. This microalga achieved a final biomass level of 55–58 g m^−2^ at concentrations of 0.09, 0.18, and 0.45 mgL^−1^ cobalt and ~45 g m^−2^ at a cobalt concentration of 4.5 mg L^−1^ (Table 4). Considering these results, the bioremediation efficiency of cobalt was studied at a cobalt concentration of 4.5 mg L^−1^, the highest concentration at which a significant biomass increase was observed. At this cobalt concentration, after two days, the removal efficiency of cobalt was ~75%, with 1.2 mg L^−1^ cobalt remaining (Figure 3, Table 4). At the end of the experiment, the removal efficiency of cobalt was 85% with an ion metal adsorption (QE formula in [24]) of 84 mg g^−1^ (Table 4).

In the same year, Li et al. [70] tested the ability of *Stichococcus bacillaris* to remove zinc, using a TW-S. *Stichococcus bacillaris* (originally isolated from an acid mine drainage in France) was tested in the presence of atmospheric air and of two different media, SWW with a zinc concentration of 2 mg L^−1^ or 3 mg L^−1^ and water from a zinc mine dump leachate (MDLW) in Braubach, Germany, with a zinc concentration of 3.3 mg L^−1^. *S. bacillaris* was tested in a TW-S in the presence of SWW in batch mode for four days at different light intensities, showing the best performance at 130 μmol photons m^−2^ s^−1^ (Figure 3). The alga exhibited a stable Zn uptake at the optimal light condition over the first 10 h, with removal efficiencies of 70% and 60% in SWW at a zinc concentration of 3 mg L^−1^ and 2 mg L^−1^, respectively. Similarly, the biomass and quantum efficiency (QE) showed higher values with higher zinc concentrations (Table 4). Then, in the same light condition, the authors tested zinc removal efficiency in a multi-TW-S setup, in which five TW-S were connected in sequence and used in a semi-continuous mode. The aim was to increase the hydraulic retention time. A higher removal efficiency was observed, because, after ten hours, higher biomass and QE values were obtained (Table 4). After this test with synthetic wastewater (SWW), *S. bacillaris* was tested in MDLW in batch mode. The removal efficiency was high in the first 4 h and then decreased slowly.

The bioremediation of heavy metals was also tested with A-T in an experiment using SWIW by Cheng et al. [43]. Cheng et al. [43] showed that SWIW was rich in high concentrations of heavy metals. The zinc concentration was around 2.8 mg L^−1^, and copper and iron concentrations were ~2 mg L^−1^. The metal concentration decreased sharply in the presence of *Chlorella pyrenoidosa* after two days, with a removal efficiency of 30–40%, and continued to decrease, albeit more slowly, until the end of the experiment (Table 4; Figure 3).

## 5. Prototype and Pilot Scale PSBRs for WW Treatment

The encouraging results obtained at laboratory scale stimulated the development of prototype or pilot scale PSBRs for bioremediation tests under more realistic field conditions. Two experiments at prototype/pilot scale using TW-S have been performed by Shi et al. [46] and González-Camejo et al. [16]

Shi et al. [46] developed a prototype TW-S with a growth surface area of 6 m^2^. A membrane pump was applied with a flow rate of 3. 8 L h^−1^ to move MWW from a storage container of 50 L to the top of the source layers, after passing a cartridge filter for removal of solid particles. For this work, Shi et al. [46] used the same strain as in the previous work [37], *Scenedesmus (Halochlorella) rubescens.* The experiment was performed from April to June in a greenhouse in Cologne, Germany, with a temperature varying between 18 °C and 32 °C and a light intensity (natural sunlight) ranging from 30 to 220 μmol photons m^−2^ s^−1^. The MWW was taken from a municipal wastewater treatment plant in Frechen (Germany) and the experiment was carried out for 32 days in the presence of four different types of MWW applied consecutively, each for 8 days in the following order: (1) MWW after secondary treatment, (2) MWW after secondary treatment with addition of phosphorus (to simulate an active phosphorus precipitation process), (3) MWW after treatment in a bio-phosphorus tank, and (4) MWW after treatment in a denitrification tank. It should be noted that, after secondary treatment ammonium was absent, after treatment in the bio-phosphorus tank phosphorus was released, and during treatment in the denitrification tank, NO_3_ was converted to N_2_. Removal efficiencies depended on the type of wastewater and nutrient but were generally >70% (Figure 4A, Table 5).

González-Camejo et al. [16] tested *Scenedesmus* sp. on a vertically oriented TW-S with a total growth surface area of ~288 m^2^ consisting of 18 modules, each with 6x 2.66 m^2^ growth area (double-sided). A pump was linked to a sand filter and a coil filter before application of the wastewater to the TW-S, and was adjusted to a flow flux of 0.5 L h^−1^. The experiment was performed for 90 days, receiving wastewater from the secondary treatment effluent of a WW treatment plant (WWTP) in the Province of Córdoba (Spain). This WWTP serves 4100 inhabitants, and its effluent is discharged into an environmentally sensitive area. Microalgae were tested in three different forms: fresh, lyophilized and stored at 4 °C. In all conditions, a good removal efficiency of different compounds (ammonium, phosphorus, total organic carbon, total carbon, inorganic carbon) was obtained (Table 5, Figure 4B). Biomass levels and removal efficiencies were influenced by inlet distance; microalgae close to the inlet showed high values of biomass (Table 6) and also higher content of nitrate and phosphorus in the biomass, because they were in contact with higher amounts of nutrients.

## 6. Utilization of Microalgal Biomass after WW Treatment

Microalgal biomass produced by WW treatment could theoretically be used for a variety of applications. In the study of Cheng et al. [43,53,54,69], production of biofuel was of central interest. It is known that cobalt stimulates the cobalt–porphyrin enzyme, accelerating the synthesis of hydrocarbons [71]. Therefore, Cheng et al. [69] monitored hydrocarbon production. They showed that, in a medium rich in cobalt, *B. braunii* produced higher amounts of hydrocarbons (up to 50% of dry weight) compared to algae grown in a control medium (~40% of dry weight). More specifically, the culture rich in cobalt produced a larger amount of hydrocarbons with C_31_ long chains (around 35% of dry weight), compared to ~15% in the control. Cheng et al. [43] showed that *C. pyrenoidosa* exhibited higher lipid content in SWW (~21% of dry weight) compared to BG11 medium (19%). Similar results were obtained by Sohail et al. [56] using *Tetradesmus obliquus* (~30% lipid content when grown with PWW) and Saleem et al. [55] using *Scenedesmus* sp.(~20% lipid content when grown with RWW).

González-Camejo et al. [16] used lyophilized biomass of *Scenedesmus* sp. from a TW-S after WW treatment as a biofertilizer. The biomass was rich in different essential elements (50.0% carbon, 5.5% nitrogen, 3.8% phosphorus, 1.6% potassium and relevant humic substances). It was mixed with vegetable compost in different concentrations and a germination test with two different plants (ryegrass and barley) was carried out. Ryegrass and barley grew better in the presence of a vegetable compost with a microalgal addition of 2% (*w*/*w*) than in a control without added microalgae. After 45 days of treatment, ryegrass achieved a dry weight of 6.4 g m^−2^ in the treated condition and 2 g m^−2^ in the control, whereas treated barley achieved a weight of 29 g m^−2^ in the treated condition vs. 10 g m^−2^ in the control.

## 7. Outlook and Perspectives

PSBRs have been successfully used to remove pollutants from various types of WW (synthetic, domestic, metal- and antibiotic-contaminated, piggery, poultry), both in laboratory and pilot plant experiments. These have revealed some advantages in comparison to suspended cultivation technology, particularly linked to cost effectiveness, water consumption and high biomass productivities and levels (see also Podola et al. [21]). PSBRs have obvious advantages for WW purification in rural areas, in small scale decentralized systems in urban areas (e.g., hospitals), and for purification of industrial WW.

However, some problems associated with the use and large scale application of microalgae in the treatment of wastewater using PSBRs remain [72]. Indeed, scaling-up requires continuous operation, including automatic harvesting of algal biomass. When natural sunlight is used for photosynthesis, relatively large areas of land are required, and this conflicts with centralized, high-throughput domestic WW facilities in urban environments. The use of algal biomass as biofertilizer, although attractive as a means to recycle nutrients, in particular phosphorus, faces the same problems as conventional sludge, namely contamination by toxicants such as heavy metals, antibiotics, hormones, pathogens, etc.

Solutions to improve the efficiency of PSBRs in the removal of pollutants from WW may involve screening a more diverse set of microalgae than has previously been employed. In this context, it should be advantageous to isolate novel algal strains from the kinds of habitats, i.e., WW, that relate to their future use in the removal of pollutants. We refer to the successful isolation of microalgal strains that utilize human urine from open “urine traps” exposed in the natural environment [52]. It could be useful to test, for example, *Cyanidiophyceae* that showed interesting results, not only in the treatment of WW in suspended cultivation but also in achieving high biomass levels in combination with phycobiliprotein production in a TW-S [73,74]. Furthermore, these organisms grow in cryptolithic conditions at very low light intensities and low pH values, requiring a smaller footprint area and being less prone to contamination [74].

Also, consortia with different microorganisms, such as bacteria and fungi, immobilized in controlled arrangements, could have interesting applications, in particular for removal of specific organic pollutants from industrial wastewaters. Indeed, metabolic cascades of different microorganisms, including algae, all immobilized in a defined sequence on PSBRs, could bind or metabolically break down various toxicants, yielding highly purified WW and algal biomass for further use as biofertilizer or extraction of valuable products.

## 8. Conclusions

PSBRs are attractive systems for WW treatment but also for algal biomass production in general, at comparatively low cost. In the future, they may replace suspended systems in several applications, if they can be properly scaled up and automated. The challenges and opportunities discussed in this short review offer a vast avenue for further research on PSBRs.

## Figures and Tables

**Figure 1 bioengineering-10-01173-f001:**
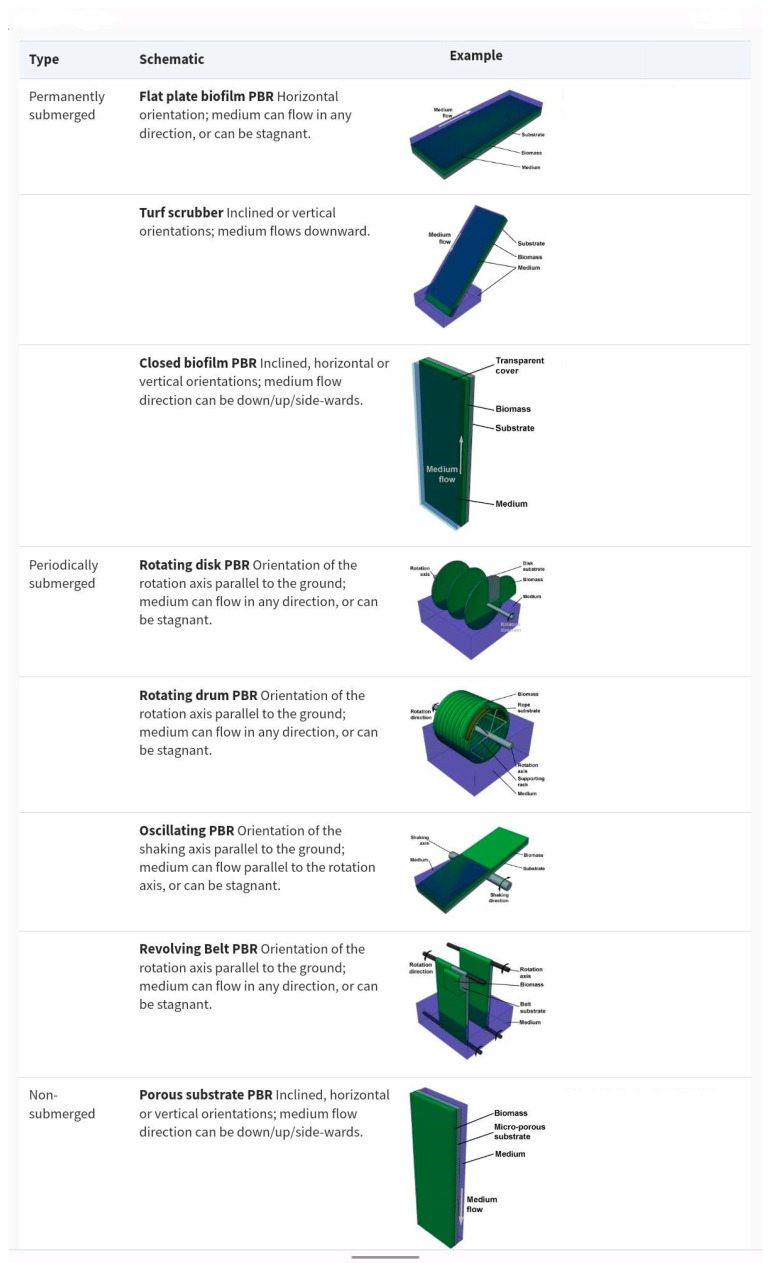
Schematic representations of different biofilm-based photobioreactors (PBRs) used in bioremediation of WW. Figure reproduced with permission from [33], and modified.

**Figure 3 bioengineering-10-01173-f003:**
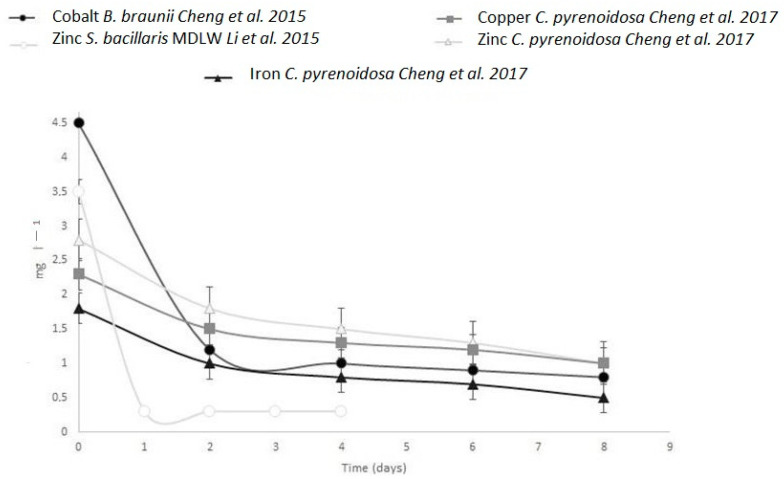
Removal trends of different metals in PSBR experiments of Cheng et al. 2017 [43], Cheng et al. 2015 [69], Li et al. 2015 [70]; other details are shown in Table 4.

**Figure 4 bioengineering-10-01173-f004:**
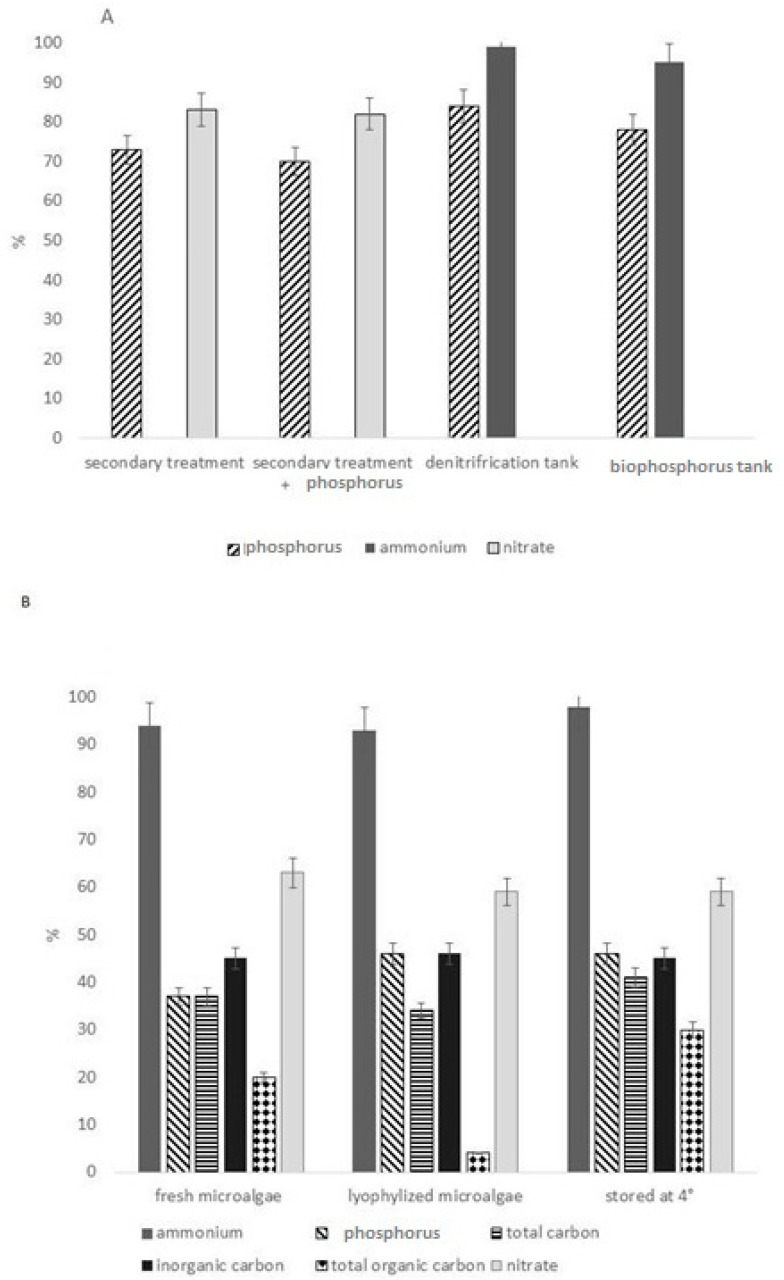
Removal efficiencies in prototype scale PSBRs: (**A**) Shi et al. [46]; (**B**) González-Camejo et al. [16].

**Table 2 bioengineering-10-01173-t002:** Microalgal growth parameters in the presence of specific wastewaters or substrates using PSBRs.

Microalgae	Initial Value (g m^−2^)	Final Value (g m^−2^)	Biomass Productivity (g m^−2^ d^−1^)	Days	Medium	PSBRs	References
*Scenedesmus (Halochlorella) rubescens*	1.4	7.7	0.9	9	SWW	TW-S	Shi et al. [37]
*Chlorella vulgaris*	1.4	11.9	1.1	9	SWW	TW-S	Shi et al. [37]
*Desmodesmus abundans*	2.5	75	8	9	Human urine	TW-S	Piltz and Melkonian [52]
*Chlorella pyrenoidosa*	8	48	3.3	8	SWIW	A-T	Cheng et al. [43]
*Scenedesmus* sp.	8	55	5.8	8	SYSWIW	A-T	Cheng et al. [53]
*Scenedesmus* sp.	8	60	6.5	8	SYSWIW + Tetracycline	A-T	Cheng et al. [53]
*Scenedesmus* sp.	8	58	6.2	8	SYSWIW + Norfloxacin	A-T	Cheng et al. [53]
*Scenedesmus* sp.	8	55	5.8	8	SYSWIW + Sulfadimidine	A-T	Cheng et al. [53]
*Chlorella* sp.	8	45	4.6	8	ADSW	A-T	Cheng et al. [54]
*Scenedesmus* sp.	5	39	5.6		RWW	TW-S	Saleem et al. [55]
*Scenedesmus* sp.	5	32	4.7		WTWW	TW-S	Saleem et al. [55]
*Tetradesmus obliquus*	3	92.83	5.13	18	PWW	TW-S	Sohail et al. [56]
*Tetradesmus obliquus*	3	87.7	4.87	18	AD	TW-S	Sohail et al. [56]
*Tetradesmus obliquus*	3	84.5	4.69	18	SWL	TW-S	Sohail et al. [56]
*Tetradesmus obliquus*	3	67.8	3.48	18	MWW	TW-S	Sohail et al. [56]
*P. maculatum*	3	15	4.2	15	PRNM	TW-S	Meril et al. [57]
*P. maculatum*	3	8	3.7	15	NM	TW-S	Meril et al. [57]
*P. maculatum*	3	5	3	15	PP	TW-S	Meril et al. [57]

**Table 3 bioengineering-10-01173-t003:** Microalgal growth parameters in specific wastewaters containing antibiotics [53] in comparison to algal growth without addition of antibiotics using PSBRs.

Microalgae	Initial Value (g m^−2^)	Final Value (g m^−2^)	Biomass Productivity (g m^−2^ d^−1^)	Days	Medium	PSBRs	References
*Scenedesmus (Halochlorella) rubescens*	1.4	7.7	0.9	9	SWW	TW-S	Shi et al. [37]
*Chlorella vulgaris*	1.4	11.9	1.1	9	SWW	TW-S	Shi et al. [37]
*Desmodesmus abundans*	2.5	75	8	9	Human urine	TW-S	Piltz and Melkonian [52]
*Chlorella pyrenoidosa*	8	48	3.3	8	SWIW	A-T	Cheng et al. [43]
*Scenedesmus* sp.	8	55	5.8	8	SYSWIW	A-T	Cheng et al. [53]
*Scenedesmus* sp.	8	60	6.5	8	SYSWIW + Tetracycline	A-T	Cheng et al. [53]
*Scenedesmus* sp.	8	58	6.2	8	SYSWIW + Norfloxacin	A-T	Cheng et al. [53]
*Scenedesmus* sp.	8	55	5.8	8	SYSWIW + Sulfadimidine	A-T	Cheng et al. [53]
*Chlorella* sp.	8	45	4.6	8	ADSW	A-T	Cheng et al. [54]

**Table 4 bioengineering-10-01173-t004:** Removal of heavy metals from wastewater using PSBRs. Heavy metal parameters: Removal efficiency is the percentage of the difference between the heavy metal concentration before and after treatment divided by the heavy metal concentration before treatment [24]. QE is the amount (mg) of metal ion adsorbed at the end of the treatment per g of dry biomass.

Microalgae	Heavy Metal	Initial Values mg L^−1^	Final Values mg L^−1^	Removal Efficiency %	QE mg g ^−1^	PSBRs	References
*Botryococcus braunii*	cobalt	4.5	0.68	85	84	A-T	Cheng et al. [69]
*Stichococcus bacillaris*	zinc	2	0.4	80	126	TW-S	Li et al. [70]
*Stichococcus bacillaris*	zinc	3	0.6	80	106	TW-S	Li et al. [70]
*Stichococcus bacillaris*	zinc	2	0.3	85	118	Multi-TW-S	Li et al. [70]
*Stichococcus bacillaris*	zinc	3	0.45	85	72	Multi-TW-S	Li et al. [70]
*Stichococcus bacillaris*	zinc	3.3	0.2	96	155	TW-S	Li et al. [70]
*Chlorella pyrenoidosa*	zinc	2.8	0.96	66	38	A-T	Cheng et al. [43]
*Chlorella pyrenoidosa*	copper	2	1	50	24	A-T	Cheng et al. [43]
*Chlorella pyrenoidosa*	iron	1.8	0.75	58	22	A-T	Cheng et al. [43]

**Table 5 bioengineering-10-01173-t005:** Removal of ammonium, phosphorus, and nitrate in MWW using a pilot scale TW-S.

		Ammonium		Phosphorus		Nitrate		
Microalgae	WW Type	Initial Values mg mL^−1^	Final Values mg mL^−1^	Initial Values mg mL^−1^	Final Values mg mL^−1^	Initial Values mg mL^−1^	Final Values mg mL^−1^	References
*Scenedesmus (Halochlorella) rubescens*	After secondary treatment	0.10	0.6	0.61	0.2	7.51	0.6	Shi et al. [46]
*Scenedesmus (Halochlorella) rubescens*	After secondary treatment and phosphorus addition	n.d		2	0.2	5.85	0.5	Shi et al. [46]
*Scenedesmus (Halochlorella) rubescens*	After secondary treatment and denitrification tank	1.79	0.8	1.95	0.3	0.52	0.3	Shi et al. [46]
*Scenedesmus (Halochlorella) rubescens*	After secondary treatment and bio-phosphorus tank	11.3	0.6	3.81	0.2	0.14	0.4	Shi et al. [46]
*Scenedesmus* sp. Fresh microalgae	After secondary treatment	24	1.4	24	1.7	24	1	González-Camejo et al. [16]
*Scenedesmus* sp. Lyophilized microalgae	After secondary treatment	5.4	3.4	5.4	2.9	5.4	2.9	González-Camejo et al. [16]
*Scenedesmus* sp. Stored at 4 °C	After secondary treatment	57	21	57	23	57	23	González-Camejo et al. [16]
		**Total carbon**		**Inorganic Carbon**		**Total Organic Carbon**		
*Scenedesmus* sp. Fresh microalgae	After secondary treatment	174	107	124	67	50	48	González-Camejo et al. [16]
*Scenedesmus* sp. Lyophilized microalgae	After secondary treatment	174	107	124	66	50	48	González-Camejo et al. [16]
*Scenedesmus* sp. Stored at 4 °C	After secondary treatment	174	107	124	67	50	48	González-Camejo et al. [16]

**Table 6 bioengineering-10-01173-t006:** Algal biomass productivities in MWW at pilot scale using a TW-S.

Microalgae	Initial Value (g m^−2^)	Final Value (g m^−2^)	Biomass Productivity (g m^−2^ d^−1^)	Days	MWW Origin	References
*Scenedesmus (Halochlorella) rubescens (western side)*	2	74	2.5	32	Frechen (Germany)	Shi et al. [46]
*Scenedesmus (Halochlorella) rubescens* *(eastern side)*	2	54	1.6	32	Frechen (Germany)	Shi et al. [46]
*Scenedesmus* sp.	3	60	0.81	90	Cordoba (Spain)	González -Camejo et al. [16]

## Data Availability

The data presented in this study are available on request from the corresponding authors.

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
