# Peer review of "Potential of Porous Substrate Bioreactors for Removal of Pollutants from Wastewater Using Microalgae"

_bioengineering, 2023, doi:10.3390/bioengineering10101173_

Round 1
Reviewer 1 Report
In this article, authors summarizes the recent research trends in porous substrate bioreactors using microalgae to treat wastewater. The subject of the article is interesting to researchers and the manuscript are well organized. However, some issues are remained to be published.
1. Many errors in punctuation (period, comma, semi-colon, etc) and word spacing are found in a whole text. It should be corrected carefully. Excessive paragraphing must be avoided in the text. Check miss-typos (NH4CL, ~45 g m-2; Table 2, Table 1; Fig.2, ammonium 78%; biomass level, and so on)
2. Some sentences can not be easily understood. I recommend to revise those sentences. “For example [15] isolated 33 different algal strains from WW and tested microalgae directly on TW-S for 12 days in the presence of secondary wastewater (SWW), showing that Scenedesmus sp. achieved the highest biomass (12 g m- 2), while cyanobacteria (Phormidium sp.) the lowest (3 g m-2).“ ”After two days, the removal efficiency of phosphorus was 42%, of ammonium 26% and of COD 15% (Fig. 2).“. ”The removal efficiency was around 93% after 4 hours; Thereafter it decreased slowly“. ”, see also Podola et al. [19].“ ”However, some problems associated with the use and large-scale application of microalgae in the treatment of wastewater remain [56]“.
3. It is necessary to describe the disadvantages of conventional photoreactors in ‘Introduction’.
4. In Fig 1, the resolution of images and the readability of letters must be improved. Also, title of fig 1 should be provided.
5. Reference notation should be uniformed ([ref number]). If you want to indicate the author, indicate the author [ref number].
6. You uses a abbreviate of SWW into two meanings of secondary wastewater (L121) and synthetic wastewater (L140). Correct it.
7. Title of table 1 should be addressed. References in all tables should be provided with reference number. New abbreviations in tables must be provided as a footnote of table.
8. From the results of Sohail et al. in table 1, 132, 119, and 118% are not removal efficiencies. They mentioned in a reference paper as “The NH4+-N removal in PWW, SWL, and AD was 132.5%, 119.4%, and 118.5% higher than MWW as depicted in Fig. 3a.”. Check and correct it.
9. Resolution of figs 2-4 should be improved. Especially, sizes of letters are too small to be read.
10. I can’t find full words of A-T (L186), please check it.
11. L230, “a light/dark cycle of 12L:12D” should be corrected into “~~ of 12:12h)”.
12. L292, one of Cheng et al. [39] should be removed.
Many errors in punctuation should be checked and corrected in a whole text. English grammar in some sentences can not be easily understood. I recommend to use English editing service to improve the quality of your work.
Author Response
The authors thank the referee for the suggestions, which were taken into acccount as indicated below and the english was reworked.
1.Many errors in punctuation (period, comma, semi-colon, etc) and word spacing are found in a whole text. It should be corrected carefully. Excessive paragraphing must be avoided in the text. Check miss-typos (NH4CL, ~45 g m-2; Table 2, Table 1; Fig.2, ammonium 78%; biomass level, and so on.
The typos were corrected in the manuscript.
2.Some sentences can not be easily understood. I recommend to revise those sentences. “For example [15] isolated 33 different algal strains from WW and tested microalgae directly on TW-S for 12 days in the presence of secondary wastewater (SWW), showing that Scenedesmus sp. achieved the highest biomass (12 g m- 2), while cyanobacteria (Phormidium sp.) the lowest (3 g m-2).“ ”After two days, the removal efficiency of phosphorus was 42%, of ammonium 26% and of COD 15% (Fig. 2).“. ”The removal efficiency was around 93% after 4 hours; Thereafter it decreased slowly“. ”, see also Podola et al. [19].“ ”However, some problems associated with the use and large-scale application of microalgae in the treatment of wastewater remain [56]“.
The sentences were modified and marked in orange color.
3. It is necessary to describe the disadvantages of conventional photoreactors in ‘Introduction’.
Done. The corrections are in yellow color.
4. In Fig 1, the resolution of images and the readability of letters must be improved. Also, title of fig 1 should be provided.
Done
5. Reference notation should be uniformed ([ref number]). If you want to indicate the author, indicate the author [ref number].
Done.
6. You uses a abbreviation of SWW into two meanings of secondary wastewater (L121) and synthetic wastewater (L140). Correct it.
Done. The corrections are in violet color.
7. Title of table 1 should be addressed. References in all tables should be provided with reference number. New abbreviations in tables must be provided as a footnote of table.
Done. The corrections are in the text.
8. From the results of Sohail et al. in table 1, 132, 119, and 118% are not removal efficiencies. They mentioned in a reference paper as “The NH4+-N removal in PWW, SWL, and AD was 132.5%, 119.4%, and 118.5% higher than MWW as depicted in Fig. 3a.”. Check and correct it.
Done. The corrections are in Orange color in the Table.
9. Resolution of figs 2-4 should be improved. Especially, sizes of letters are too small to be read.
Done.
10. I can’t find full words of A-T (L186), please check it.
Done. The corrections are in light blue in the text.
11. L230, “a light/dark cycle of 12L:12D” should be corrected into “~~ of 12:12h)”.
Done. The correction is in grey color
12. L292, one of Cheng et al. [39] should be removed.
Done.
Reviewer 2 Report
This paper shows that removing the pollutants from wastewater via using microalgae on potential of porous substrate bioreactors. The mechanisms about the processes of removing pollutants from wastewater by microalgae on potential of porous substrate bioreactors were demonstrated in the context. The studies were quite systematic and the resulted were well organized by the authors. I’d like to recommend the publication of this paper in bioengineering after revision.
1. The structure of the porous structure should be explained in Table 1 by authors.
2. The bioreactors duration should be discussed in the context by authors.
3. The major factors and parameters for high heavy metal removal efficiency should be discussed in the context by authers.
Author Response
The authors thank the referee for the suggestions, which were taken into account as addressed below
1. The structure of the porous structure should be explained in Table 1 by authors.
Done the corrections are in light violet color.
2. The bioreactors duration should be discussed in the context by authors
Done. The corrections are in Red color in the text.
3. The major factors and parameters for high heavy metal removal efficiency should be discussed in the context by authers.
Done the correction are in Red color in the text.
Round 2
Reviewer 1 Report
Authors tried to address a manuscript to follow reviwer's comments and suggestions. A few minor revision is required.
- It is recommended to remove the keyword of ‘environmental biotechnology’ as it is too broad, I suggest to add ‘pollutants removal’.
- Clearly explain what the TOC, TC, and IC in the 10th row of Table 5 mean.
- Correct NH4CL in line 155.
Author Response
The authors thank referee and all suggestion are taken in account.